# Environmental Conditions in Middle Eastern Megacities: A Comparative Spatiotemporal Analysis Using Remote Sensing Time Series

Shahin Mohammadi [1], Mohsen Saber [2,*], Saeid Amini [3], Mir Abolfazl Mostafavi [4,5], Gavin McArdle [6] and Hamidreza Rabiei-Dastjerdi [7,8]

1 Department of Remote Sensing and GIS, Faculty of Earth Sciences, Shahid Chamran University of Ahvaz, Ahvaz 61357-43136, Iran
2 School of Surveying and Geospatial Engineering, College of Engineering, University of Tehran, Tehran 14174-66191, Iran
3 Department of Surveying and Geomatics Engineering, University of Isfahan, Isfahan 81746-73441, Iran
4 Center for Research in Geospatial Data and Intelligence, Université Laval, Quebec City, QC G1V 0A6, Canada
5 Center for Interdisciplinary Research in Rehabilitation and Social Integration, Laval University, Quebec City, QC G1V 0A6, Canada
6 School of Computer Science and CeADAR, University College Dublin (UCD), D04 V1W8 Dublin, Ireland
7 School of Architecture, Planning, and Environmental Policy & CeADAR, University College Dublin (UCD), D04 V1W8 Dublin, Ireland
8 Social Determinants of Health Research Center, Isfahan University of Medical Sciences, Isfahan 81746-73461, Iran
* Correspondence: m.saber@ut.ac.ir

**Abstract:** Rapid and timely evaluation and monitoring of the urban environment has gained significant importance in understanding the state of urban sustainability in metropolises. Multi-source remote sensing (RS) data are a valuable source for a comprehensive understanding of urban environmental changes in developing countries. However, in the Middle East, a region with several developing countries, limited study has been conducted to understand urban environmental changes. In this study, to evaluate the changes in the urban environment, 32 metropolises in the Middle East were studied between 2000 and 2019. For this purpose, a comprehensive environmental index (CEI) integrated with Google Earth Engine (GEE) platform for processing and analysis is introduced. The results show degraded environmental conditions in 19 metropolises based on a significant increasing trend in the time series of the CEI index. The highest increasing trend in the value of the CEI was observed in the cities of Makkah, Jeddah, Basra, Riyadh, and Sana'a. The results also show that the percentage of urban areas in all 32 cities that falls into the degraded class varies from 5% to 75% between 2005 and 2018. The results of CEI changes in megacities, such as Ajman, Tehran, Jeddah, Makkah, Riyadh, Karaj, and Sana'a show that these cities have increasingly suffered from the degradation of environmental conditions since 2001. According to the results, it is recommended to pay more attention to environmental issues regarding the future of urban development in these cities. The proposed approach in this study can be implemented for environmental assessment in other regions.

**Keywords:** urban environmental conditions; trend analysis; big data; megacity; Middle East

## 1. Introduction

More than half of the world's population was settled in urban areas by 2007 [1]. The most significant population growth in the 21st century occurred in cities. According to predictions, the urban population is expected to further increase significantly [2], reaching more than 6 billion people by 2050 [3]. Due to this significant growth in urban areas, the management of cities, especially big cities and metropolises, has become a complex issue and a significant challenge [4].

The excessive concentration of population and complex interactions in urban areas increases pressure on the urban environment. The formation of heat islands, an increase in air pollution, a reduction of natural areas as well as unequal urban development are some of the most known destructive effects of excessive population growth in developing societies, such as Middle East countries [5]. Due to the importance of monitoring the quality of the urban environment, this issue has become one of the key topics in the field of human and environmental studies [6]. Due to this sensitivity, the issues of the urban environment and quality of human life have become one of the most critical issues of the 21st century [7]. For this reason, in recent years, assessing the quality of the urban environment has gained significant importance in urban management planning [8]. Today's conditions in cities have pushed experts in different directions to assess the quality of cities and plan to improve their quality [9].

Acquiring information about environmental conditions can be achieved using ground measurement stations. However, there are some limitations in this method of data gathering. On the one hand, the process of measuring these parameters is complicated, and the creation and maintenance of these stations are costly and time-consuming [10–12]. In the past years, remote sensing (RS) has become an efficient tool in spatial modelling and an essential resource in monitoring the Earth. Processing these data provides essential information to users, and for this reason, in the last two decades, experts have conducted much research to process these data [12–14].

Today, various indicators have been presented on different subjects, including urban quality. With the help of these indicators and conditions, other regions of the world can be assessed [15,16]. By using these indicators, it is possible to evaluate the environment around us [10,14,17,18]. The Comprehensive Environmental Index (CEI) is one of the most well-known indicators in this field [19]. Ref [20] used this index and evaluated the state of the urban environment in 17 big cities (mainly in East Asia). In another study, Ref [21] conducted a study for developing an environmental assessment index. The index introduced in this research was calculated using several RS indices using the Google Earth Engine (GEE) platform.

The review of articles in this field shows that various studies have been conducted to investigate the environmental degradation of cities in some parts of Asia. Nevertheless, a comprehensive evaluation has not been performed to investigate the environmental changes in the metropolises of the Middle East. In addition, the review of previous studies showed that in a limited number of cases, the evaluation of the urban environment degradation status had been performed by relying solely on RS data [18,22]. In addition, the processing of remote sensing data for time series analysis is time-consuming and requires a high volume of calculations. Providing an optimal solution to speed up processing will increase the efficiency of the research. This makes it easier to analyse the environmental condition of a region. Therefore, in this study, the GEE platform and satellite products are used to calculate the CEI index. This index has been evaluated for 32 metropolises in the Middle East between 2000 and 2019. The proposed approach is presented based on the GEE platform and can be easily developed and used in subsequent studies by other researchers for other regions of the world. The steps of this research are summarised in four general sections. The study area and the research steps are described in Section 2. The results of this research are presented in Section 3. The results are analysed and discussed in Section 4. Finally, we present some conclusions in Section 5.

## 2. Materials and Methods

### 2.1. Case Study

The Middle East includes the southern and eastern borderlands of the Mediterranean Sea, the Arabian Peninsula, Iran, and North Africa (Figure 1). This area covers approximately seven million square kilometres, and about 370 million people live in this region. The Middle East has diverse topographic, geological and climatic conditions and vegetation cover (VC), and these conditions have created various soils in this region [23]. The maxi-

mum height of the Middle East is Mount Damavand in Iran, and its approximate height is 5671 m. In general, the climate in the Middle East is dry and semi-arid. The land cover (LC) in this region is highly diverse and is spread from complex mountainous areas to desert ones. Most of the Middle East has a hot and dry climate with barren regions. However, several large and important rivers, such as the Nile, Tigris, Euphrates and Karun rivers, are also in this region. The existence of these rivers has contributed to the sustainability of agriculture and the emergence of great civilisations in this region. The Middle East is considered an important energy source for the world due to its vast oil reserves. The Middle East is a strategic region regarding geography and energy, and 56.6% of the world's oil resources are in this region. Saudi Arabia, Iran, Iraq, Kuwait, and the United Arab Emirates are among the countries that have the most significant oil resources in the world. This area also has vast reserves of natural gas. Among the countries of the Middle East, Iran and Qatar have the second and third largest sources of natural gas reserves in the world, respectively.

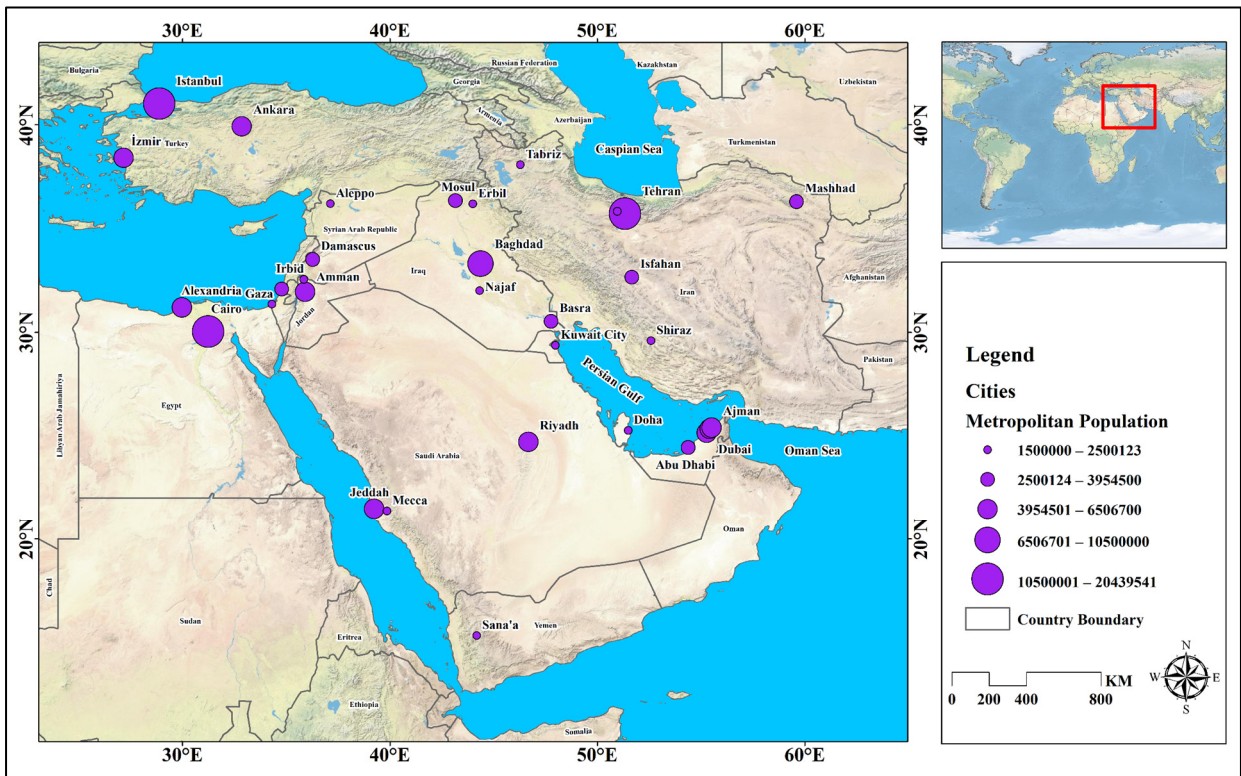

**Figure 1.** Physiographical map of the study area (Middle East).

*2.2. Data Sets*

In this study, three data sets are used to investigate and monitor the spatial–temporal pattern of urban environmental changes in 32 metropolises in the Middle East (Table 1). These data sets are the concentration of particles smaller than 2.5 microns ($PM_{2.5}$), land surface temperature (LST), and Normalised Differential Vegetation Index (NDVI) data. Population information and the boundaries of urban areas were used as additional data sets. In addition, the GEE platform has been used to perform analysis and evaluation. The location and size of the cities studied in this research are shown in Figure 1.

**Table 1.** List of selected 35 megacities in the Middle East in this study (www.worldatlas.com) (accessed on 16 September 2022).

| Rank | City | Country | Metropolitan (Population) | City (Population) |
|---|---|---|---|---|
| 1 | Cairo | Egypt | 20,439,541 | 9,500,000 |
| 2 | Tehran | Iran | 16,672,000 | 9,134,000 |
| 3 | Istanbul | Turkey | 15,519,267 | 15,241,177 |
| 4 | Baghdad | Iraq | 10,500,000 | 8,126,755 |
| 5 | Riyadh | Saudi Arabia | 6,506,700 | 6,506,700 |
| 6 | Ankara | Turkey | 5,663,322 | 5,067,565 |
| 7 | Dubai | United Arab Emirates | 5,640,000 | 3,287,007 |
| 8 | Sharjah | United Arab Emirates | 5,640,000 | 1,405,000 |
| 9 | Ajman | United Arab Emirates | 5,640,000 | 490,035 |
| 10 | Alexandria | Egypt | 4,984,387 | 4,984,387 |
| 11 | Amman | Jordan | 4,642,000 | 4,061,150 |
| 12 | İzmir | Turkey | 4,394,694 | 5,067,565 |
| 13 | Jeddah | Saudi Arabia | 4,276,000 | 4,276,000 |
| 14 | Tel Aviv | Israel | 3,954,500 | 438,818 |
| 15 | Mosul | Iraq | 3,750,000 | 1,683,000 |
| 16 | Mashhad | Iran | 3,600,650 | 3,372,090 |
| 17 | Isfahan | Iran | 2,989,070 | 2,000,000 |
| 18 | Damascus | Syria | 2,900,000 | 2,078,000 |
| 19 | Abu Dhabi | United Arab Emirates | 2,784,490 | 2,784,490 |
| 20 | Basra | Iraq | 2,750,000 | 2,750,000 |
| 21 | Tabriz | Iran | 2,500,123 | 2,000,000 |
| 22 | Doha | Qatar | 2,382,000 | 1,850,000 |
| 23 | Kuwait | Kuwait | 2,380,000 | 2,380,000 |
| 24 | Sana'a | Yemen | 2,167,000 | 1,937,451 |
| 25 | Irbid | Jordan | 2,050,300 | 582,276 |
| 26 | Gaza | Palestine | 2,047,969 | 590,481 |
| 27 | Mecca | Saudi Arabia | 2,042,000 | 2,042,000 |
| 28 | Karaj | Iran | 1,967,000 | 1,967,000 |
| 29 | Shiraz | Iran | 1,869,001 | 1,869,001 |
| 30 | Aleppo | Syria | 1,800,000 | 1,800,000 |
| 31 | Erbil | Iraq | 1,750,564 | 1,750,564 |
| 32 | Najaf | Iraq | 1,500,000 | 1,389,500 |

In this research, we used the $PM_{2.5}$ data set based on the data introduced by the atmospheric composition analysis group at Washington University in St. Louis [24]. This data set contains the annual average concentration of $PM_{2.5}$, and its spatial resolution is 1 km. This data set uses the combination of aerial optical depth (AOD) from MODIS, MISR, and SeaWiFS sensors and uses the GEOS-Chem chemical transport model to estimate its final product. This data set was calibrated using Geographically Weighted Regression with global ground-based $PM_{2.5}$ observations [25]. These data are available as a product, and to receive it, you can refer to the archive of the NASA Socio-Economic Data and Applications Center (SEDAC) website. More details about this data set can be found at: https://sedac.ciesin.columbia.edu/ (accessed on 16 September 2022). In recent years, this $PM_{2.5}$ data source has attracted the attention of researchers and has been used in several studies around the world [26–29]. The final data set is released after performing some preprocessing on the data. This data set can easily be imported into different online platforms, such as the GEE platform.

MO(Y)D11A1 is one of the most well-known LST data sets. This product is provided by processing MODIS sensor data. The MODIS sensor is installed on Terra and Aqua satellites. The spatial resolution of this product is 1 km. This product has been available in the data archive and distribution system of the National Aeronautics and Space Organization since 2000 [30]. LST data can be used to investigate heat islands. The intensity of the urban heat island generally has a daily cycle and reaches its maximum during the night [31]. This issue has been investigated in recent studies. According to these studies, the maximum

intensity of the urban heat island in most Asian cities occurs in the late afternoon, night or early morning [32]. For this reason, in this research, night LST data have been used to analyse urban environmental changes. Temperature values that were higher than 60 °C and lower than −60 °C were considered erroneous and excluded from this research. Then, the annual mean LST for each metropolis in the GEE environment was extracted.

In this research, the 16-day vegetation index products (MOD13Q1) with a resolution of 250 m from the United States Geological Survey's (USGS) Distributed Earth Process Active Archive Center were used. We used the GEE platform to calculate annual mean NDVI values based on a 16-day composite data time series. To extract the cities' boundaries, each city's border was determined using Google Maps, and then, the urban limits of all the metropolises in 2019 were drawn. In Table 2, a list of the data used in this study is fully stated.

**Table 2.** List of the data used in this study.

| Indicator | Data Generation | Country | Temporal Scale | Provider |
|---|---|---|---|---|
| $PM_{2.5}$ | MODIS + MISR + SeaWiFS + GEOS-Chem | 1000 m | 2000–2019 | SEDAC |
| LST | Terra/MODIS/Night/Daily | 1000 m | 2000–2019 | NASA |
| NDVI | Terra/MODIS/NDVI/16 day | 250 m | 2000–2019 | NASA |
| City Boundaries | Google Maps | - | - | - |

*2.3. Methodology*

In this research, $PM_{2.5}$, LST, and NDVI data were used to evaluate and monitor the spatiotemporal pattern of urban environmental changes. For this purpose, we used the GEE platform to analyse the metropolises of the Middle East. Google Maps was used to extract urban areas. After preparing these layers, the CEI was calculated to evaluate urban environmental changes. To calculate this index, the combination of the values of $PM_{2.5}$, LST, and NDVI data is used. To evaluate urban environmental changes from 2001 to 2019, the average environmental conditions in 2000 in all metropolitan cities were considered the base state. In this research, the Mann–Kendall (MK) test method is used to analyse the trend of urban environmental changes. In addition, we calculated the correlation between the standard deviation of each parameter and the standard deviation of CEI to investigate the influence of different parameters on CEI. The workflow of urban environmental assessment using CEI is shown in Figure 2.

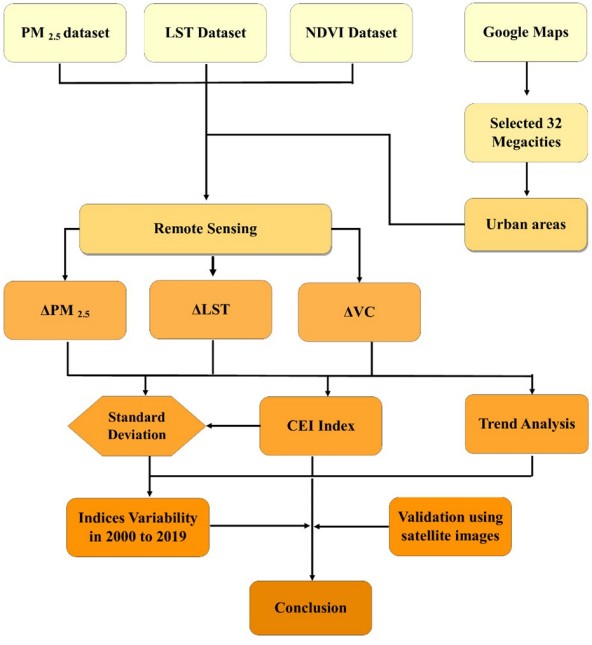

**Figure 2.** The workflow of urban environmental assessment using CEI.

2.3.1. CEI Derivation

As stated in this study, we used CEI to assess the urban environmental changes in the metropolises of the Middle East. This index is calculated from the combination of changes in the three parameters: LST, VC, and PM$_{2.5}$. The geometric mean value of CEI can be calculated using LST, VC, and PM$_{2.5}$ values, as mentioned in Equation (1) [19].

$$\text{CEI}_i^k = \sqrt[3]{\left(\Delta \text{PM}_i^k + 1\right) * \left(\Delta \text{LST}_i^k + 1\right) * \left(\Delta \text{VC}_i^k + 1\right)} \tag{1}$$

Here, CEI$_i^k$ is the environmental changes in pixel i and year k (2001, 2002, ... , 2019). The changes in this index are in the range from 1 to 101. The larger values of this index indicate the greater degree of destruction in the urban environment. In this equation, $\Delta \text{PM}_i^k$, $\Delta \text{LST}_i^k$, and $\Delta \text{VC}_i^k$ are the changes in the concentration of PM$_{2.5}$, LST, and VC, respectively, in pixel i and from 2000 to 2019 with normalized values. Because the two parameters of the LST and the concentration of PM$_{2.5}$ have a direct relationship with the destruction of the environment, Equation (2) was used to normalize these indicators:

$$\Delta V_i^k = \frac{\left(V_i^K - V_i^{2000}\right) - \text{min}_v}{\text{max}_v - \text{min}_v} * 100 \tag{2}$$

In Equation (2), $\Delta V_i^k$ is the normalized change for the values of PM$_{2.5}$ and LST parameters from 2000 to 2019. $V_i^{2000}$ and $V_i^K$, respectively, indicate the concentration of PM$_{2.5}$ microns and the temperature of the land surface at pixel i in the year 2000 and year K (k ranges from 2001 to 2019). min$_v$ and max$_v$ are the minimum and maximum changes in PM$_{2.5}$ concentration and LST from 2001 to 2019, respectively. Then, the VC in each pixel is calculated using Equation (3) [33]:

$$\text{VC}_i = (N_i - N_{\text{min}})/(N_{\text{max}} - N_{\text{min}}) \tag{3}$$

In Equation (3), VC$_i$ is the value of VC in pixel i, N$_i$ is the value of NDVI in pixel i, and N$_{\text{min}}$ and N$_{\text{max}}$ are the minimum and maximum NDVI values, respectively. VC has an inverse relationship with other parameters and the environmental index. Therefore, Equation (4) has been used to normalize this index ($\Delta \text{VC}_i$):

$$\Delta \text{VC}_i^k = \frac{\text{max}_{vc} - \left(\text{VC}_i^K - \text{VC}_i^{2000}\right)}{\text{max}_{vc} - \text{min}_{vc}} * 100 \tag{4}$$

In Equation (4), VC$_i^{2000}$ and VC$_i^K$ show the VC values of pixel i in the year 2000 and year K (2001 to 2019). In addition, min$_{vc}$ and max$_{vc}$ parameters are the minimum and maximum VC changes from 2001 to 2019, respectively. In addition, examining the temporal anomaly of the CEI index can play a significant role in evaluating the environmental condition of cities. To further investigate the anomaly in the time series of the CEI, the anomaly of the CEI (ACEI) was calculated according to Equation (5):

$$\text{ACEI}_i = \sqrt{\frac{1}{n} \sum_{k=1}^n \left(\text{CEI}_i^k - \overline{\text{CEI}_i}\right)^2} \tag{5}$$

where CEI$_i^k$ is the CEI value at pixel i in the year K. $\overline{\text{CEI}_i}$ is the average CEI at pixel i and n is the total number of years. This index is the average of the CEI annual time series from 2001 to 2019 at the pixel level. In Equation (5), CEI$_i^k$ is the CEI value at pixel i in year K.

The average and standard deviation of CEI were used to investigate the variability of the state of destruction in urban areas [34]. Based on the CEI values, standard deviation and average values of CEI for metropolitan cities were extracted between 2001 and 2019. According to the results, cities were grouped into five categories (Table 3). These five groups

are: improved, moderately improved, no change, moderately degraded, and degraded. Based on the classification results at the pixel level for each metropolis, the percentages of area for each type of environmental change in the urban area were calculated from 2001 to 2019.

**Table 3.** Categories of environmental change in the Middle East.

| Type | Improved | Moderately Improved | No Change | Moderately Degraded | Degraded |
|------|----------|---------------------|-----------|---------------------|----------|
| Classification | $<\mu - 1.5\delta$ | $\mu - 1.5\delta$ to $\mu - 0.5\delta$ | $\mu - 0.5\delta$ to $\mu + 0.5\delta$ | $\mu + 0.5\delta$ to $\mu + 1.5\delta$ | $>\mu + 1.5\delta$ |
| CEI | 36.30 | 36.30 to 41.56 | 41.56 to 46.83 | 46.83 to 52.1 | 52.1< |

### 2.3.2. Trend Analysis

In this study, the MK trend analysis was used to analyse the trend of the CEI index and its influencing factors. The annual average value of LST, VC, PM$_{2.5}$ and CEI for each urban area is calculated to extract the time series of data points in Middle East megacities from 2000 to 2019. To detect the significance of changing trends, Zs and *p* value statistics were used as indicators of the time series trends for each city. The standard normal test of Zs is calculated using Equation (6):

$$Z_s = \begin{cases} \frac{S-1}{\sqrt{Var(S)}} & S > 0 \\ 0 & S = 0 \\ \frac{S+1}{\sqrt{Var(S)}} & S > 0 \end{cases} \tag{6}$$

In Equation (6), positive and negative values of Z$_S$ indicate increasing and decreasing trends in this test, respectively. A significant downward or upward trend in the CEI time series will exist if the null hypothesis is rejected at a particular significance level. We used the Python programming language in this study to analyse the MK test.

## 3. Results

### 3.1. Inter-Annual Variation and Trends in PM$_{2.5}$, LST, and VC

In this study, the annual changes of LST, VC, PM$_{2.5}$ and CEI concentrations in 32 metropolises in the Middle East are investigated. The results show that according to PM$_{2.5}$, the highest average yearly amount of air pollution belongs to the Doha metropolis (Figure 3), where the annual average amount of air pollution is 59.1 µg m$^{-3}$. The cities of Abu Dhabi and Kuwait are ranked next in terms of air pollution in the study area, with annual average values of 54.2 and 54 µg m$^{-3}$, respectively. The cities of Istanbul, Aleppo, and Irbid are among the less polluted metropolises with an annual pollution amount of less than 20 µg m$^{-3}$. According to the statistics reported by the World Health Organisation (WHO), these cities are considered clean cities [35]. The trend of PM$_{2.5}$ concentration changes showed that at a significance level of 1%, the cities of Jeddah, Makkah and Sana'a, and at a significance level of 5%, the cities of Tehran, Irbid, Jeddah, Karaj, Makkah, Izmir, and Sana'a have a significant trend of increasing pollution (Table 4). On the other hand, the cities of Alexandria and Cairo, at a significance level of 5%, have a downward trend in the amount of PM$_{2.5}$ concentration. In addition, the results of this study showed that in the cities of Jeddah, and Makkah, the average annual growth is about 1 µg m$^{-3}$ (Table 4).

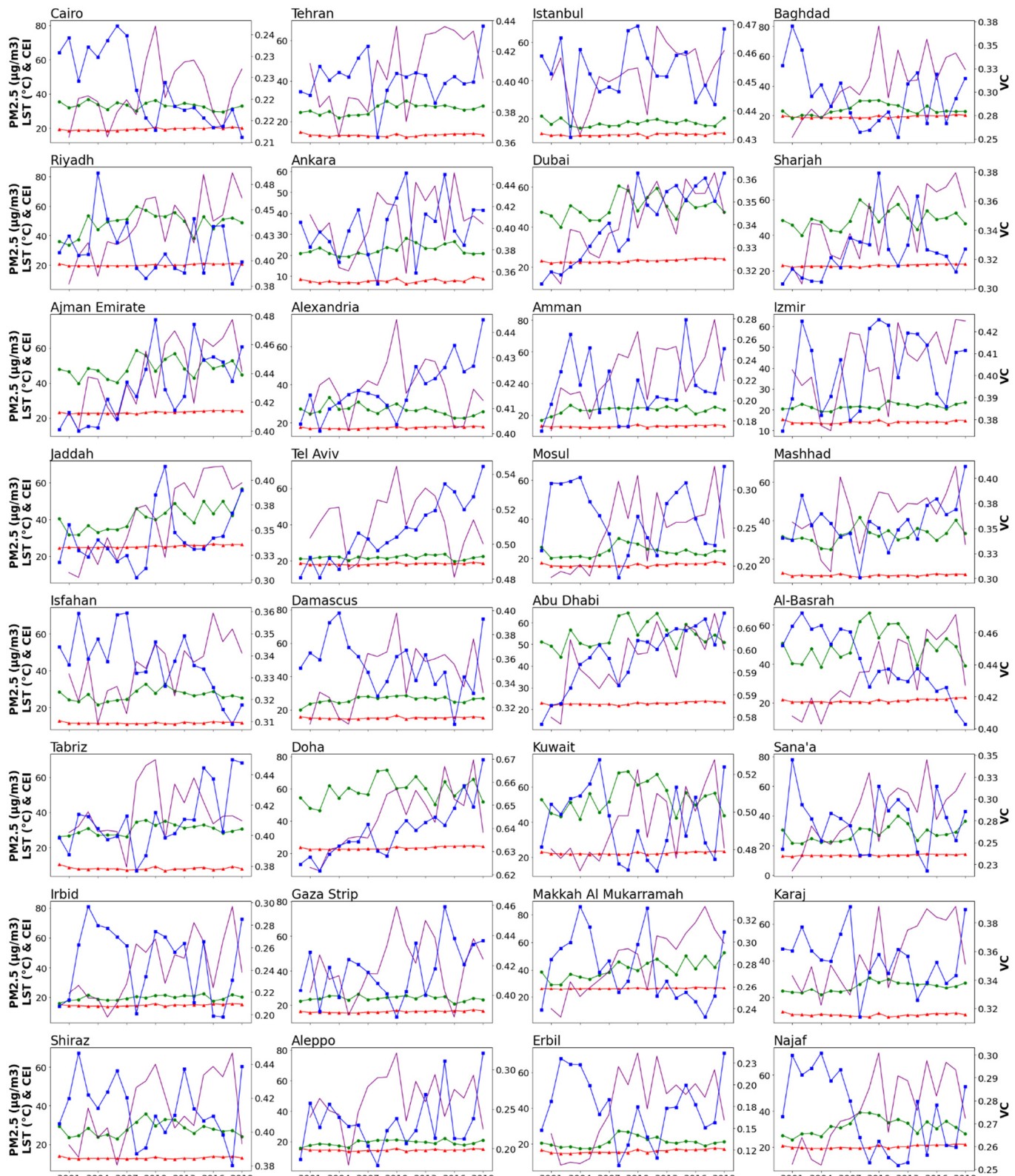

**Figure 3.** This Interannual changes in LST, VC and PM$_{2.5}$ concentrations in the study area from 2000 to 2019 ( ▬■▬ VC, ▬●▬ PM2.5, ▬▲▬ LST, and ▬ CEI).

**Table 4.** MK trend analysis in LST, PM$_{2.5}$ concentration, VC and CEI in the selected Middle East megacities from 2000 to 2019.

| City Name | LST | PM$_{2.5}$ | VC | CEI | City Name | LST | PM$_{2.5}$ | VC | CEI |
|---|---|---|---|---|---|---|---|---|---|
| Cairo | 0.102 ** | −0.139 * | −0.001 ** | 1.165 | Isfahan | 0.012 | 0.103 | −0.002 ** | 1.864 ** |
| Tehran | 0.031 | 0.207 * | 0 | 1.999 * | Damascus | 0.03 * | 0.145 | −0.002 | 1.533 |
| Istanbul | 0.061 * | 0.044 | 0 | 1.249 * | Abu Dhabi | 0.081 ** | 0.212 | 0.001 ** | 1.843 ** |
| Baghdad | 0.08 * | 0.163 | −0.002 | 2.618 ** | Al−Basrah | 0.107 ** | 0.19 | −0.003 ** | 2.863 ** |
| Riyadh | 0.087 ** | 0.41 | −0.001 | 2.853 ** | Tabriz | −0.008 | 0.171 | 0.002 * | 0.391 |
| Ankara | 0.05 | 0.122 | 0.001 | 1.063 | Doha | 0.118 ** | 0.404 | 0.002 ** | 2.843 ** |
| Dubai | 0.123 ** | 0.293 | 0.003 ** | 2.248 ** | Kuwait | 0.068 | 0.229 | 0 | 2.618 ** |
| Sharjah | 0.088 ** | 0.23 | 0.001 | 2.437 ** | Sana'a | 0.069 ** | 0.445 ** | 0 | 2.852 ** |
| Ajman | 0.103 ** | 0.219 | 0.003 ** | 2.396 ** | Irbid | 0.068 ** | 0.199 * | −0.001 | 2.689 ** |
| Alexandria | 0.062 ** | −0.174 * | 0.001 ** | 0.034 | Gaza | 0.070 ** | 0.003 | 0.001 | 0.849 |
| Amman | 0.065 ** | 0.162 | 0.001 | 1.960 * | Makkah | 0.057 ** | 0.929 ** | −0.002 | 3.833 ** |
| Izmir | 0.043 | 0.124 * | 0.001 | 1.676 ** | Karaj | 0.016 | 0.265 * | −0.001 | 2.335 ** |
| Jeddah | 0.101 ** | 0.919 ** | 0.002 * | 3.231 ** | Shiraz | 0.019 | 0.08 | −0.001 | 1.925 * |
| Tel Aviv | 0.049 * | 0.046 | 0.003 ** | 0.37 | Aleppo | 0.021 | 0.135 | 0.001 | 0.673 |
| Mosul | 0.08 * | 0.132 | −0.001 | 2.012 ** | Erbil | 0.141 ** | 0.072 | −0.001 | 2.388 * |
| Mashhad | 0.036 | 0.253 | 0.002 | 1.013 | Najaf | 0.112 ** | 0.241 | −0.002 * | 2.780 ** |

** Significant at the 1% level. * Significant at the 5% level.

The analysis of temperature changes showed that most of the cities in the Middle East (23 cities) have a significant increasing trend at a substantial level of 5%. Meanwhile, in 17 cities, at a significance level of 1%, a growing trend in temperature was observed. These cities are Abu Dhabi, Alexandria, Cairo, Amman, Dubai, Gaza, Basra, Sharjah, Ajman, Doha, Erbil, Irbid, Jeddah, Mecca, Najaf, Riyadh, and Sana'a. These results show that the highest temperature increase trend is in the cities of Erbil, Dubai, and Doha. The average annual temperature of these cities increased by 0.14, 0.12, and 0.11 degrees Celsius, respectively. Additionally, the results showed that the cities of Makkah and Jeddah have the highest average night temperatures in the year at 27 and 26 degrees Celsius, respectively. On one hand, the evaluation results of the VCI showed that dense VC (VC => 0.6) can be seen in the urban area of Abu Dhabi and Doha. On the other hand, in the cities of Erbil and Amman, the amount of VC is very thin and poor (VC < 0.3). The trend analysis of this parameter during the studied period also showed a significant increases in urban areas at the significance level of 1% in the cities of Abu Dhabi, Alexandria, Dubai, Ajman, Doha and Tel Aviv, and at the significance level of 5% in the cities of Jeddah and Tabriz. This considerable increase indicates an improvement in environmental conditions in these cities from the point of view of the VC parameter. However, in Cairo, Basra, and Isfahan, a significant decrease in the level of 1 or 5 per cent in the amount of VC can be seen (Figure 3).

In Figure 4, a normalised map of VC, PM$_{2.5}$, and LST with the names ΔVC, ΔPM$_{2.5}$, and ΔLST is presented. The general evaluation of the VC index according to the ΔVC parameter shows environmental degradation in the central parts of the Middle East, including Iraq, Egypt, and Syria, which is an alarming issue. However, the investigation of ΔLST shows that this parameter often has acute conditions and high temperatures in the southeastern and northeastern parts of Iran (provinces of Sistan Baluchistan and North and South Khorasan), Dubai, Oman, and Muscat (countries around the Oman Sea). The interpretation of the ΔPM$_{2.5}$ index shows that severe air pollution is visible in the central parts of the Middle East. The results of this index also show the highest level of environmental destruction from the point of view of air pollution occurring in the central parts of the Middle East. The results are similar to the VC.

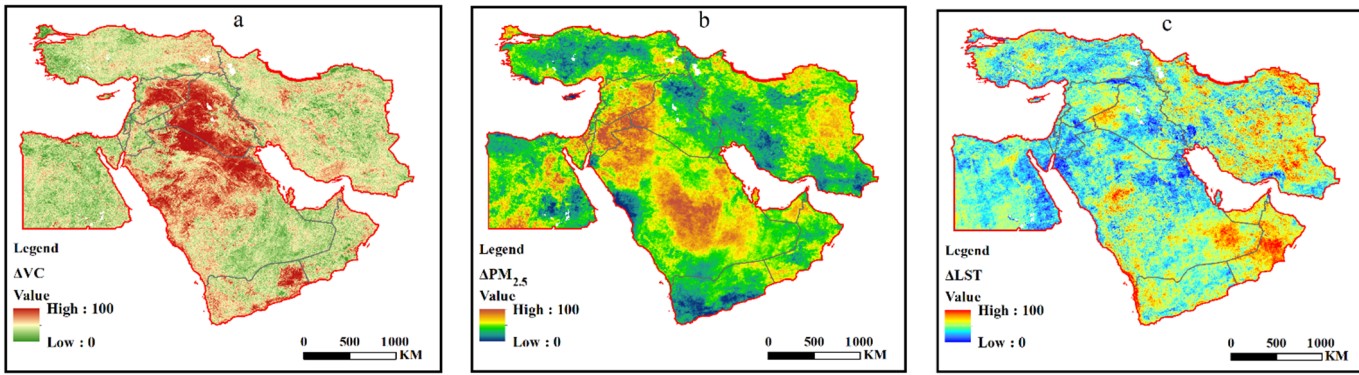

**Figure 4.** Map of three normalized environmental indicators in the study area from 2000 to 2019. (**a**) ΔVC (**b**) ΔPM$_{2.5}$ (**c**) ΔLST.

### *3.2. Inter-Annual Variation and Trends in CEI*

Analysis of the CEI trend shows that 25 of the 32 cities in the study had a significant increasing trend at a significance level of 5%. However, only 19 metropolitan cities showed a significant increase at a significance level of 1%. No substantial decreasing trend was observed in any of the cities in the study (Table 2). The most increasing trends in the CEI value can be seen in Makkah, Jeddah, Basra, Riyadh, and Sana'a. These cities are in acute conditions in terms of environmental issues, and the environmental quality in the urban areas of these cities decreased between 2001 and 2019 (Figure 3). According to the CEI, there is no significant reducing or increasing trend in Alexandria, Cairo, Tel Aviv, and Tabriz. The quality of environmental issues in these areas has not undergone visible changes. In Figure 5, a box plot presents descriptive statistics and the annual average CEI from 2001 to 2019. The maximum long-term average of the CEI (from 2001 to 2019) is observed in the cities of Aleppo and Makkah. The results indicate that the environmental conditions in these areas are poor. In addition to these cities, the cities of Riyadh and Amman, Tehran and Sharjah also show high CEI values. The lowest level of CEI was observed in Basra, Mosul, and Ankara. Interquartile ranges are high in the cities of Ajman, Basra, Jeddah, Mecca, Tehran, and Sharjah. This shows a great diversity in the environmental conditions in the studied metropolises during the study period. In addition, Figure 5 shows a skewness in most of the studied cities. This deviation indicates a lack of stability in environmental conditions in the Middle East. Examples of cities exhibiting this instability are Dubai, Karaj, and Basra.

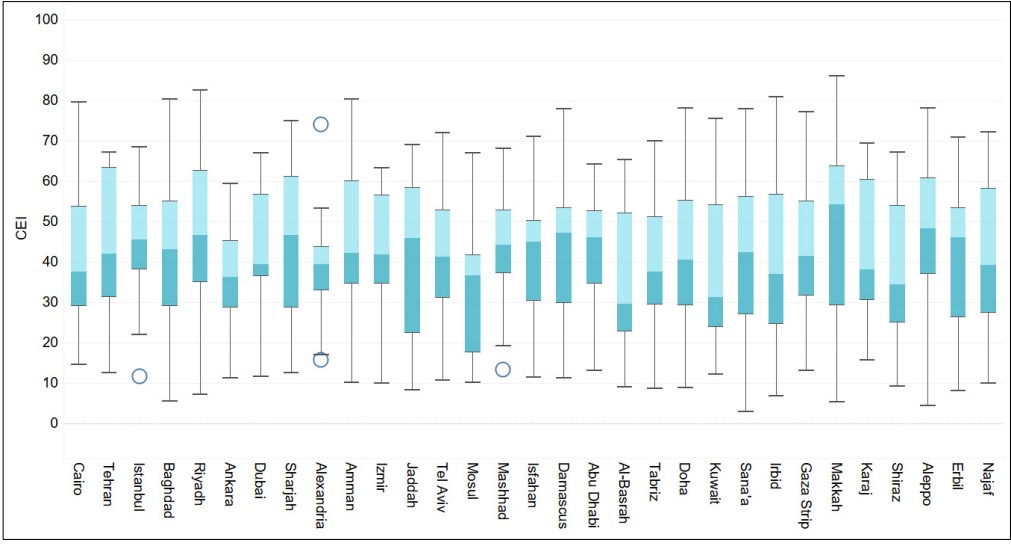

**Figure 5.** Box plot of CEI in Middle East metropolises.

To demonstrate the spatial pattern of environmental conditions in the urban areas of the Middle East, the average CEI was extracted between 2001 and 2019. The average CEI results were categorised into eight classes according to Figure 6. The results of this section show that a large share of the cities of Riyadh, Sharjah, Amman, Sana'a, and Makkah is in the degraded class, with a value of more than 53. Apart from Amman (the capital of Jordan), only the southern part of the Middle East suffered severe degradation. This degradation covered the entire urban area of cities, such as Riyadh, Sharjah, Sana'a, Makkah, and Aleppo. This shows that in the newly developed urban areas in a city such as Amman, around the urban nuclei, degradation is very evident. This is while in other cities, the concentration and development in the entire urban area are impressive. Some degraded environmental areas can be seen in cities such as Ajman, Tehran, Mashhad, Isfahan, Abu Dhabi, Doha, Gaza, and Najaf. However, degradation in these cities is not very high. In addition, in Mosul, Basra, Kuwait, and Irbid, the majority of the urban area is in the class of low degradation.

CEI variability represents the yearly difference between CEIs during the study period. For the assessment of CEI variability, we used an index known as Abnormality in CEI (ACEI). According to Figure 7, there is a severe anomaly in the CEI in Makkah, Sana'a, Doha, Jeddah, Ajman, Riyadh, and Tehran. This anomaly may be due to the climatic, economic, or social conditions of these areas in different periods. According to the results, the abnormality of this index is not only related to the population. This justification is because there was not much diversity in cities with large populations, such as Ankara and Baghdad. Still, in metropolises with a smaller population, such as Karaj, the amount of this diversity was high. In Ankara, Alexandria, Aleppo, Mosul, and Tabriz, the levels of anomaly and diversity are low. This shows relatively stable environmental conditions in these cities. As shown in Table 3, we investigated the classification of annual CEI changes in urban areas under different environmental conditions. The results of the investigation are presented in Figure 8. The results show that the degraded class for urban areas has ranged from 5% in 2005 to about 75% in 2018. In addition, the percentage of urban areas with unchanged conditions varied between 5% in 2005 and 15% in 2008. This shows that the areas of the no-change class have not changed significantly. The results of this study show that the highest environmental degradation was in 2018. During the studied period, it always has a downward slope (increasing the area of areas with higher CEI values). The range of improved sections (class with minimum destruction) has decreased.

Figure 8 shows interannual changes for urban areas in each metropolis under different environmental conditions. Since 2001, urban areas in cities such as Ajman, Tehran, Jeddah, Makkah, Riyadh, Karaj, and Sana'a have suffered from deteriorating environmental conditions. In other words, many of these cities have experienced unfavourable conditions in the last few years. According to the results of this research, there is no significant improvement in the environmental situation in the investigated cities. In addition, among these cities, the percentage of improved areas in Tel Aviv and Alexandria has increased in recent years. However, during the studied period, no significant improvement in the CEI was observed for them.

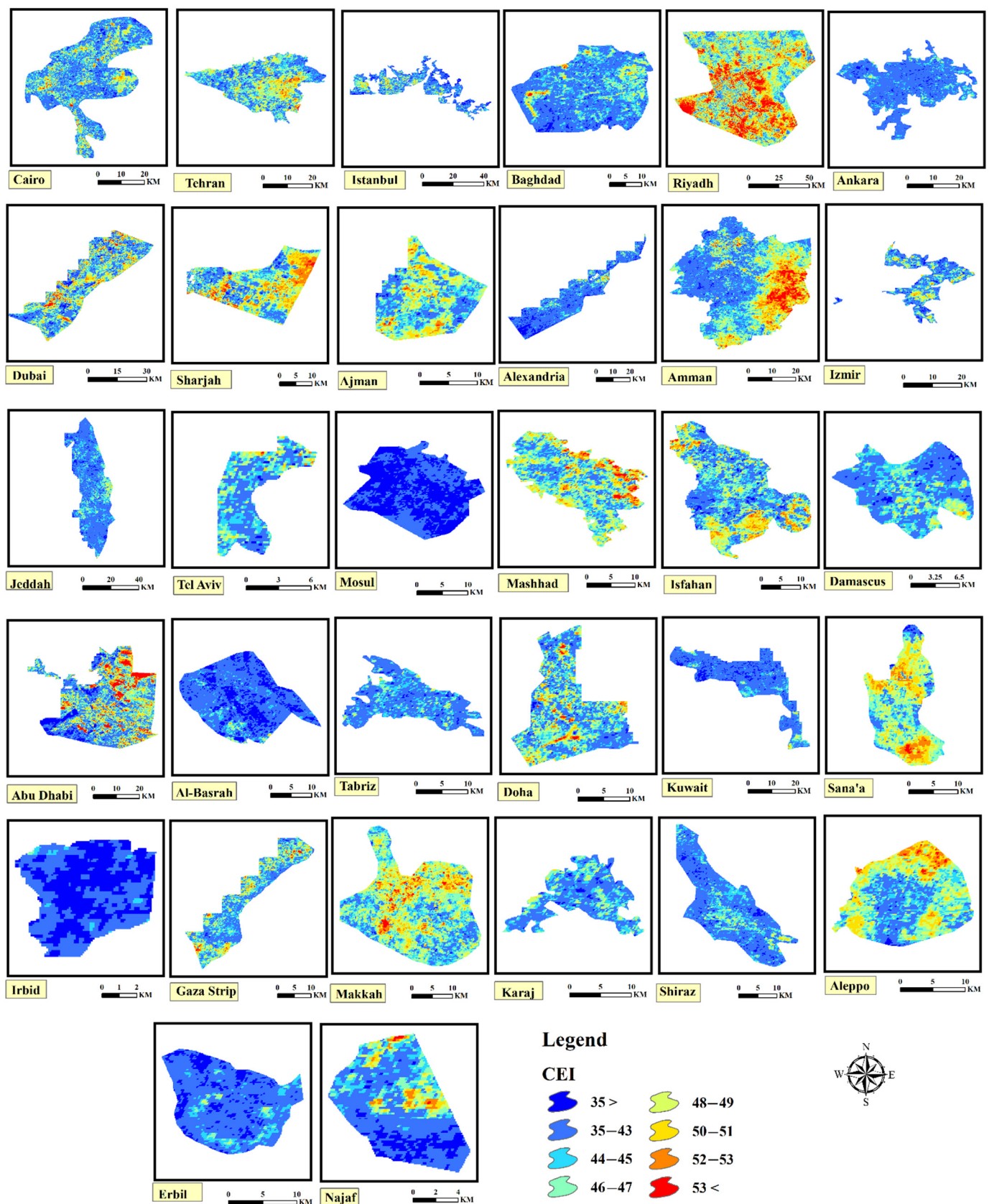

**Figure 6.** CEI changes in the urban areas over the study area (from 2001 to 2019).

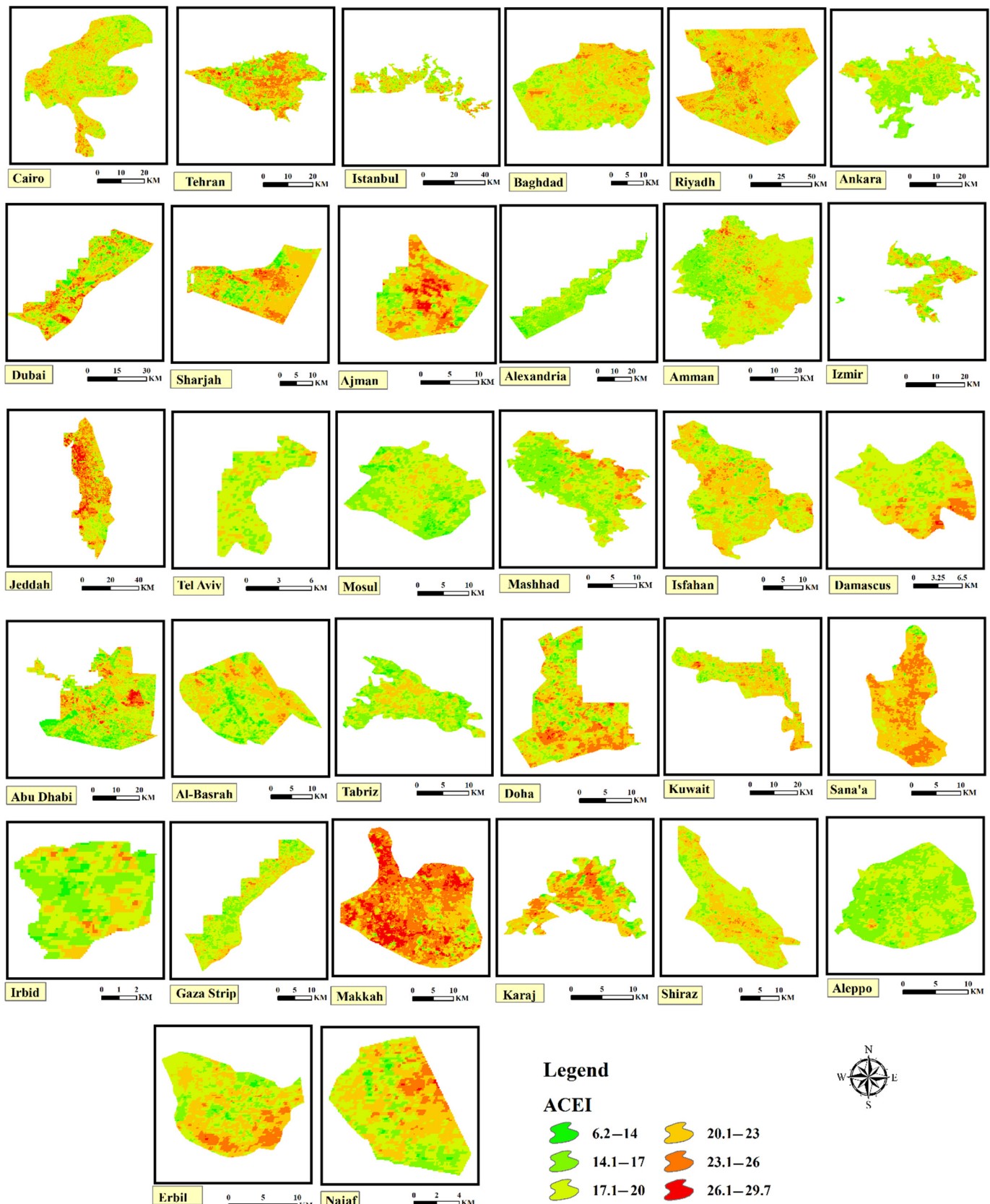

**Figure 7.** ACEI changes in urban areas over the study area (from 2001 to 2019).

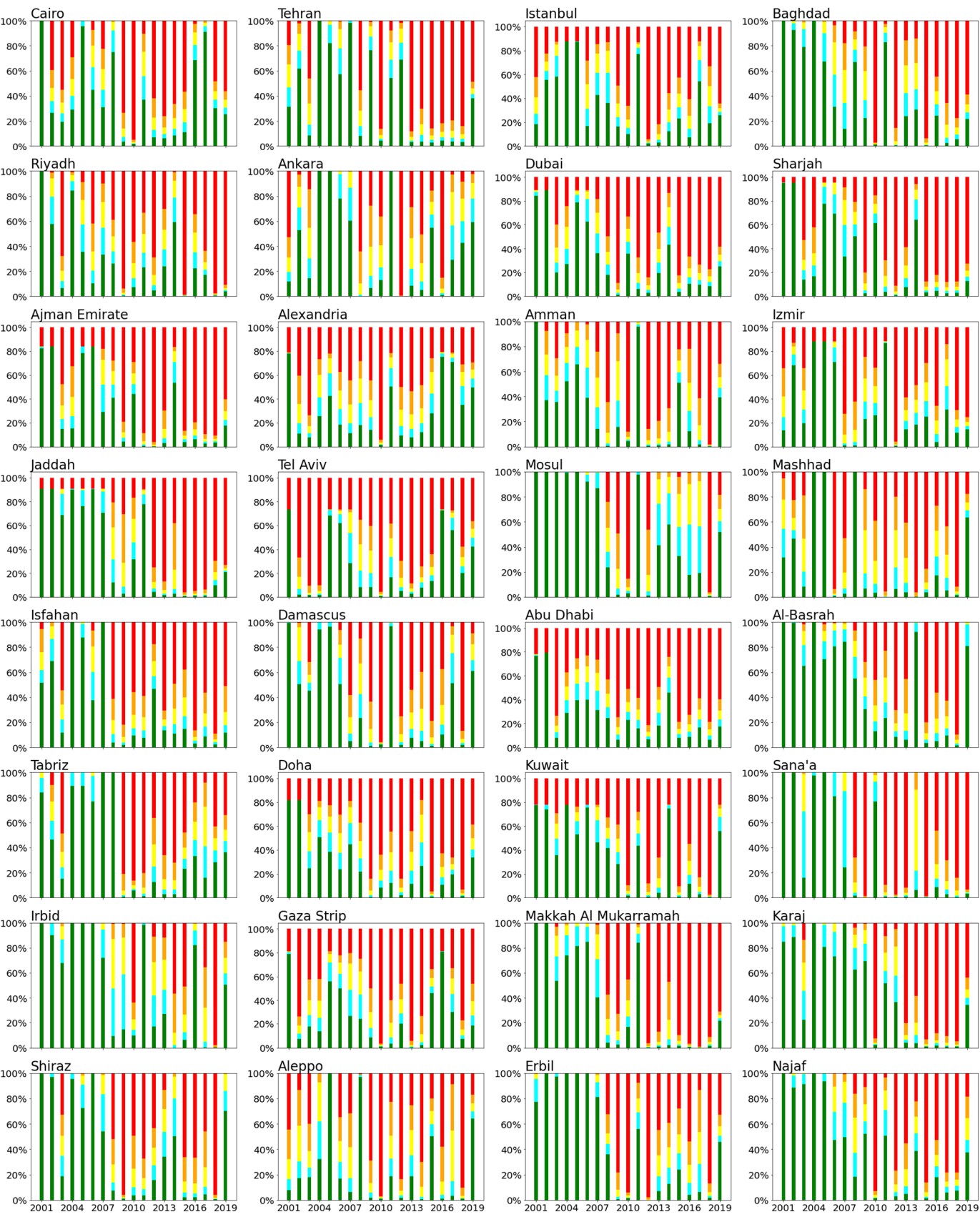

**Figure 8.** The rate of annual changes under different environmental conditions in urban areas in the study area from 2001 to 2019. (The classes are: ◼ degraded, ◼ moderately degraded, ◻ no change, ◼ moderately improved, ◼ improved).

Figure 9 shows that the per cent of residential areas with a significant increase in CEI was identified in each metropolis based on MK trend analysis. For this purpose, the significant trend of each pixel was investigated in the long term. The results of this analysis show that the highest increase in the significance levels of 1% and 5% occurred in Makkah, Sana'a, and Jeddah. This increase in these cities was 92, 91 and 89 per cent of the entire urban area, respectively. At the significance level of 1% and 5%, the least changes in an increasing trend were observed in the cities of Tel Aviv, Alexandria, and Aleppo. The long-term annual average rate of Aleppo city for the CEI was very high, but the increasing percentage and the changing trend of its area are low. This shows that the amount of abnormality in this city was low (Table 4). The information obtained from the ACEI also confirms this issue. In cities such as Makkah, Jeddah, Riyadh, and Karaj, the amount of variation of the CEI during the study period is high. In other words, in the dominant part of the urban area (more than 70% of the entire area) at the pixel level, there is a significant increasing trend of CEI at the levels of 1% and 5%.

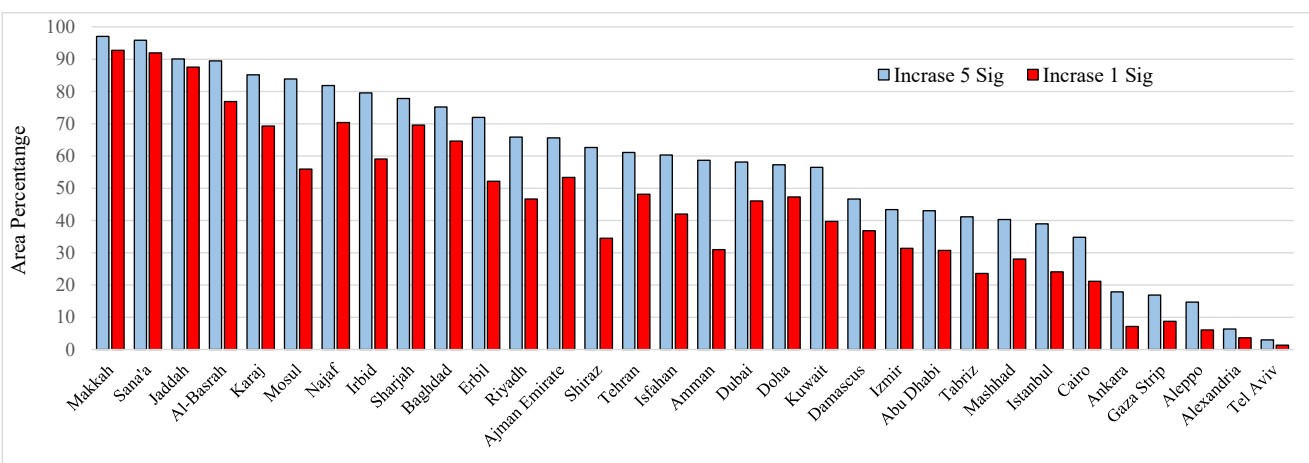

**Figure 9.** The percentage of urban areas has a trend at a significance level of 1 and 5 per cent for the cities in the study area.

### 3.3. Regression Analysis between ACEI Abnormality and Normalized Indices

In this section, the regression analysis results between ACEI and long-term anomaly of ΔVC, ΔLST, and ΔPM$_{2.5}$ parameters are investigated (Figure 10). The analysis shows which variable has a greater impact on the pattern of spatial changes of the CEI in the long term. According to the results, in most of the investigated cities, there is the highest correlation between the two indexes of spatial diversity in CEI and ΔVC. This shows that this index greatly impacts the state of environmental degradation. According to the results, Irbid, Erbil, Basra, and Karaj cities have the highest degree of correlation. In other words, the changing trend of CEI in these cities is more similar to the changing trend of the VC parameter, and there is a similar pattern of spatial changes in these cities between these two components (ΔVC and ACEI). In some cities, including Baghdad, Sharjah, Najaf, Ajman and Isfahan, there is a higher correlation between the ACEI and the ΔLST anomaly index. The lowest correlation was observed between spatial changes of ACEI and ΔPM$_{2.5}$ anomaly. Nevertheless, there is a positive correlation between the spatial pattern of the two mentioned indicators in Sharjah, Basra, and Mashhad. Assessing the spatial changes of the degree of correlation shows that in cities such as Irbid, Tel Aviv, Damascus, Gaza, Cairo, Alexandria, Amman, and Izmir, which are often located in the vicinity of each other and the west of the Middle East, the CEI has a significant impact on the VC of urban areas. Nevertheless, in the Persian Gulf and Oman Sea coastal cities, such as Doha, Ajman, Dubai, and Sharjah, the main correlation of CEI anomaly is caused by the LST parameter.

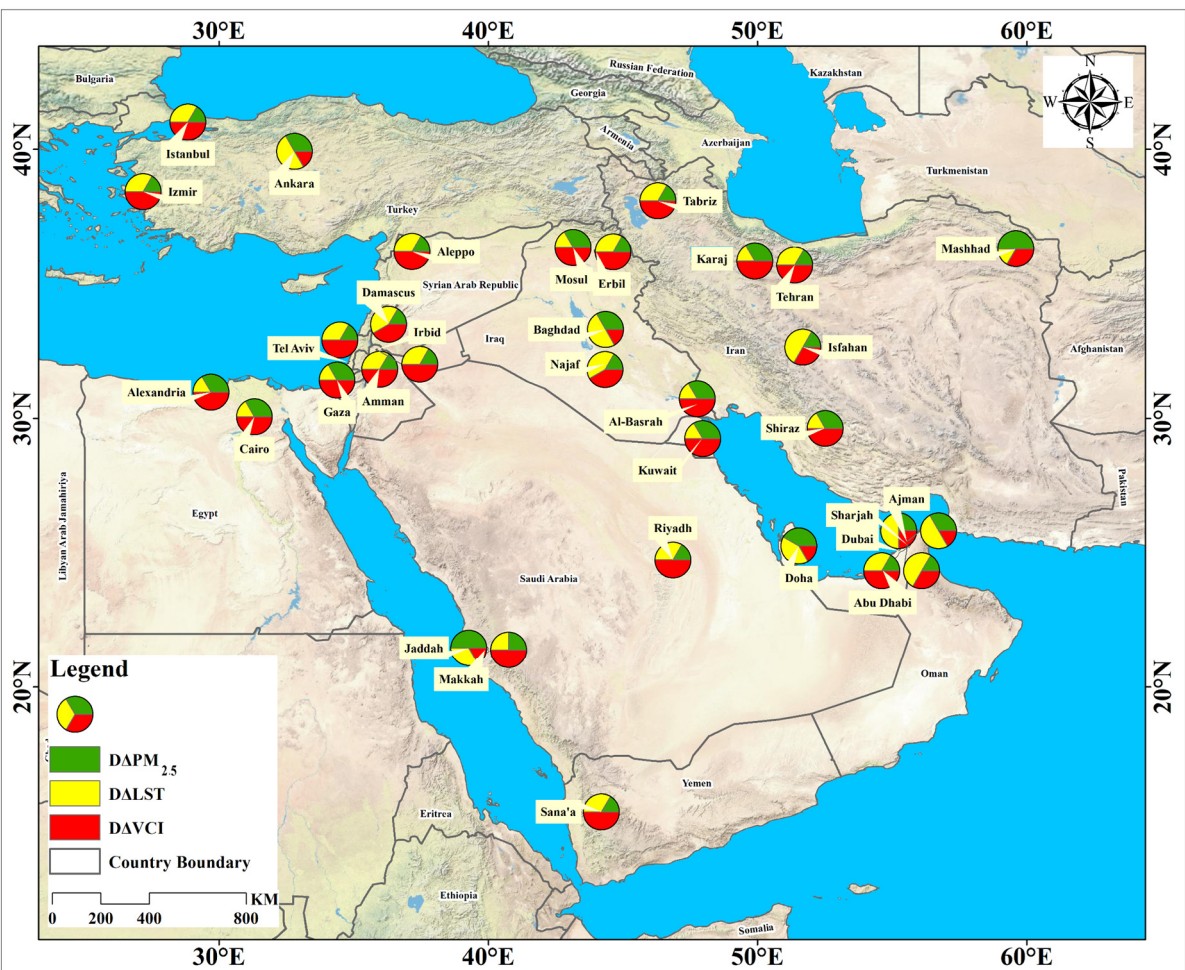

**Figure 10.** Prioritization map of modelling parameters in determining CEI anomaly in the study area.

## 4. Discussion

In this study, several different sources of remote sensing data are used to perform a complete assessment of urban environmental changes from 2000 to 2019 in selected metropolises of the Middle East. The results showed that at the significance level of 5 and 1 per cent, the concentration of $PM_{2.5}$ significantly increased in eight and three cities, respectively. A review of the studies conducted using satellite and ground data of $PM_{2.5}$ shows a significant annual increase in some countries in the Asian continent, especially in the Middle East, such as Iran, Bahrain, Saudi Arabia, and Turkey [36,37]. Research work by [38] was conducted to investigate the role of the $PM_{2.5}$ index in mortality rates. It has been shown that high levels of $PM_{2.5}$ had an influential role in mortality rates in the last two decades in several Middle Eastern countries. Other studies have shown that urbanisation increases the concentration of $PM_{2.5}$, and there is a direct relationship between the increase in population density and the concentration of $PM_{2.5}$ [39]. According to the urban growth and development of different regions of the Middle East, essential planning is required every year. If there is no principled and codified plan to reduce pollution in big cities, in the coming years, the concentration of this pollutant will increase, especially in residential areas, and it will reach a critical level.

The trend analysis of the results at the significance level of 1 and 5 per cent showed that Cairo and Basra had a decreasing VC trend. However, there is a growing trend with VC in Abu Dhabi, Alexandria, Dubai, Ajman, Doha and Tel Aviv. The study of [40] showed that in the metropolis of Isfahan from 1995 to 2019, the urban area has grown at high speed, and on the other hand, the VC class of the area has decreased. This phenomenon has been confirmed in similar research [41]. According to a study conducted by [42] in

Cairo, the increase and growth of the population caused the loss of areas with VC in this region. In addition, ref [43] modelled the rate of VC change. This study predicted that 14% more of Cairo's VC will become urban areas by 2025. According to our research, there is a slight change in the VC cover in Istanbul from 2000 to 2019. These results are consistent with the analysis conducted by [44] and [20]. VC plays an influential role in the city environment. The presence of VC in the urban environment helps to regulate the weather, purify water and air, reduce temperature and reduce the effectiveness of heat islands. Many researchers have investigated this issue, and the results of these studies confirm this issue [45–47]. For this purpose, increasing the abundance and level of VC in urban areas can be a strategy to moderate the UHI (urban heat island) effect and reduce cities' air pollution and environmental degradation.

According to our research, at the significance level of 1 and 5 per cent, there is a significant increasing trend in the amount of LST in the metropolitan cities of Abu Dhabi, Alexandria, Cairo, Amman, Dubai, Gaza, Basra, Sharjah, Ajman, Doha, Erbil, Irbid, Irbid, Jeddah, Mecca, Najaf, Riyadh, and Sanaa. In Iranian metropolises, including Tehran, Karaj, and Isfahan, urban development has caused heat islands [48]. However, our study showed no significant increase in the annual average temperature of the built-up areas in these cities. These results may indicate that the yearly average temperature is not a valid candidate to show the severity of urban heat islands (UHI). The study of [49] stated that a significant increasing trend is observed in the nightly data of the LST from 2003 to 2014. However, Ref [50] indicated that in the last decade, in the urban limits of Cairo, the temperature has increased by an average of 1.5 degrees Celsius.

In this research, the assessment of the annual changes in environmental conditions was investigated based on the analysis of CEI during the statistical period. The results showed environmental degradation in all the metropolises of the Middle East. In some cities, such as Tel Aviv, Alexandria, Ankara, and Tabriz, there was no significant trend in the degradation of environmental conditions. Nevertheless, a significant upward trend in the CEI can be seen in 19 of the 32 cities at the 1 and 5 per cent levels. Some studies show that some countries, such as India and Bangladesh, have a similar pattern to the cities of the Middle East. Nevertheless, an improvement in the environmental conditions has been observed in China. This issue could be due to the preparation of the plan to take preventive measures and control air pollution, which was approved by the Chinese government in 2013 [20].

Dubai, Mecca, Tehran, and Ajman were chosen to evaluate the accuracy of the CEI abnormality results. These cities have shown drastic changes during the studied period. The results of this investigation are presented in Figure 11. For this purpose, the CEI changes in extreme class ranges were investigated by comparing high-resolution satellite images in 2000 and 2020. The evaluation of this index shows that urban growth has been very fast in all four cities. In addition, the results show that the increase in population density and urbanization is quite clear. Reference [51] stated that Dubai is one of the megacities that has seen significant urban development and growth in the last two decades (Figure 11a). This factor has caused the temperature in urban areas to be more than 2.5 degrees warmer than the surrounding areas, and heat islands have been created in parts of the urban area. In another study, [52] stated that urban growth in Dubai would reduce green space and VC; consequently, this issue will threaten urban health. Reference [53] stated that in the last 30 years in Saudi Arabia, there have been significant changes in land use and land cover. These changes are due to the solid national development path that the government of this country started during this period, and the large national oil revenues have had a significant impact on it. With this growth of urban development, the level of residential areas in Mecca has increased by 174% from 1986 to 2013 (Figure 11b). A part of Tehran city is investigated in Figure 11c. The changes in construction and urban development density are evident. Changes in the density and urban development in this region have caused anomalies in the CEI. According to Table 4, significant changes in air pollution and LST can be seen in Tehran. These parameters have caused this city to suffer a

high rate of destruction with a rapidly increasing trend. The research of [54] stated that the trend of increasing temperature and decreasing VC cover in Tehran in the last two decades is evident. According to the results of this research, in Tehran, the horizontal growth of the city is associated with a decrease in the extent of VC. In addition, the increase in temperature and decrease in VC, indiscriminate constructions and the instability of green space in the metropolis of Tehran affect the sustainability of the environment. For this purpose, providing appropriate solutions is required.

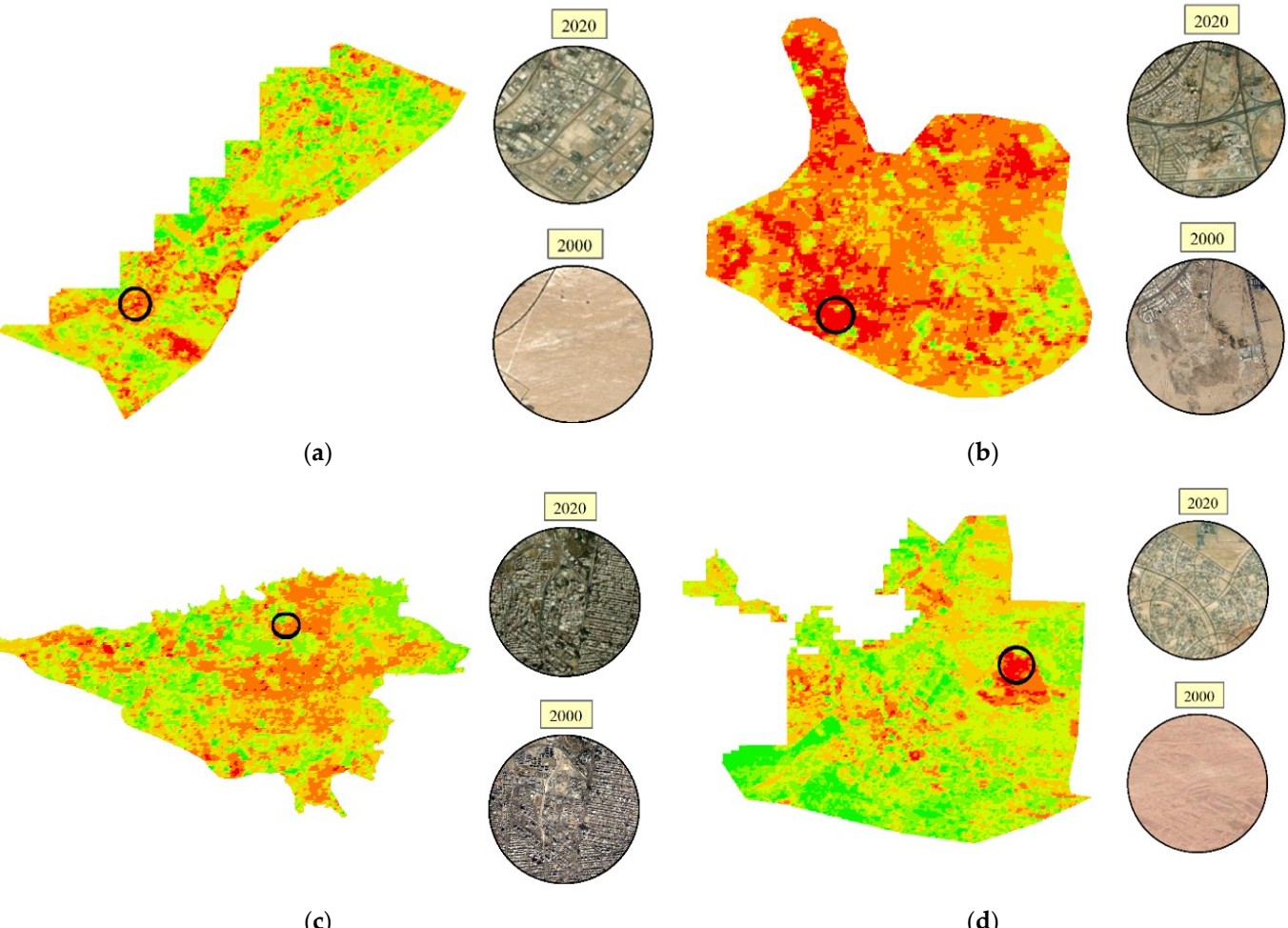

**Figure 11.** Map Analysis of satellite images of four cities in the Middle East with severe anomalies in the CEI (from 2000 to 2019). (**a**) Dubai, (**b**) Mecca, (**c**) Tehran, (**d**) Abu Dhabi.

Some parts of Abu Dhabi showed that the environmental anomaly is very severe (Figure 11d). The investigation of this anomaly indicates that in this city, in the last two decades, there has been no basic planning for the development and growth of the urban area with the aim of sustainable environmental development. According to the research results of [54], the urban area of Abu Dhabi has increased by more than 500% from 1993 to 2013. The rapid growth and status of the CEI in the urban area of Abu Dhabi show that the city has grown at high speed, but there is no planning to maintain the quality of the environment in this city.

Although there is uncertainty in the modelling, the results show that the environmental conditions in the urban environments of the Middle East are unsuitable, and these unfavourable conditions are increasing. For this purpose and to reduce the risks related to health that occur due to the deterioration of air quality in developed urban areas, administrative measures should be taken to control the spread of air pollution. The comparison of

this research's results with previous research findings shows the effectiveness of satellite products for monitoring and continuous monitoring of the urban environment on large geographical scales. Compared to previous studies, newer products have been used along with a more extended study period. Thirty-two metropolises in the Middle East were selected, and some of the limitations in previous studies were resolved. Due to the use of the powerful GEE platform in this study, it is possible to perform calculations quickly and automatically. Using the codes developed in this platform, obtaining similar results can be achieved worldwide.

## 5. Conclusions

This research provides a practical approach to investigate the spatiotemporal patterns of urban environmental change in Middle Eastern metropolises by performing trend analyses of multi-source RS data. The proposed approach was implemented using the GEE platform to creates an automatic and fast approach for environmental monitoring based on the CEI index. This platform provides the possibility to easily process a large volume of RS data that has many computational complexities. This platform provides online access to various sources of RS data, and compared to other technical software, it does not have the limitations of downloading and storing data locally. In addition, through the GEE platform, it is possible to apply various algorithms to perform the required pre-processing of the satellite data, and these operations are performed easily and at high speed in this platform. The proposed approach can be used at a high speed on a large scale to monitor and evaluate the environmental sustainability situation in different parts of the world and can be easily developed. The analysis of changes and trends of CEI showed that the environmental conditions have degraded substantially at a significance level of 1% and 5% in 19 metropolises of the Middle East. The most degradation occurred in Mecca, Jeddah and Basra, respectively. Meanwhile, the lowest level of environmental degradation can be seen in Alexandria, Tel Aviv and Tabriz. In cities such as Ajman, Tehran, Jeddah, Makkah, Riyadh, Karaj, and Sana'a since 2001, the growth of the cities has been high, and the environmental quality in these urban areas has had high environmental degradation. In other words, an extensive area of these cities has had environmental degradation conditions in the last few years. This research provides valuable concepts for planners and managers specializing in urban planning. Using these concepts during policy implementation, appropriate solutions can be delivered to reduce the resources that threaten the urban environment. This research can facilitate sustainable urban development. In future studies, it is suggested to use satellite data with a higher spatial resolution (such as the Landsat satellite set) and to assess a more extended period. In addition, it is suggested that more parameters are used to model the environmental assessment, and this approach will lead to more comprehensive results. The source codes for the proposed algorithms can be used in other research works, and the access link is included in this article.

**Author Contributions:** Conceptualisation, S.M., H.R.-D., M.S., S.A., G.M. and M.A.M.; methodology, S.M., H.R.-D., M.S. and S.A.; software, M.S. and S.M.; validation, M.S. and S.M.; formal analysis, S.M., H.R.-D., M.S. and S.A.; investigation, S.M., H.R.-D., M.S. and S.A.; data curation, S.M., H.R.-D., M.S. and S.A. writing—original draft preparation, S.M. and S.A.; writing—review and editing, S.A., S.M., H.R.-D., M.S., G.M. and M.A.M.; visualisation, M.S.; supervision, H.R.-D.; project administration, H.R.-D. and S.M. All authors have read and agreed to the published version of the manuscript.

**Funding:** This research received no external funding.

**Data Availability Statement:** The GEE repository and the view from the results are publicly available at: https://code.earthengine.google.com/?accept_repo=users/mohsensaberms/CEI (accessed on 16 September 2022) and https://mohsensaberms.users.earthengine.app/view/cei-middleeast (accessed on 16 September 2022).

**Acknowledgments:** The authors would like to thank net friends from gis.stackexchange.com (accessed on 16 September 2022) for the GEE operation's generously professional instruction.

**Conflicts of Interest:** The authors declare no conflict of interest.

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
