# Peer review of "Environmental Conditions in Middle Eastern Megacities: A Comparative Spatiotemporal Analysis Using Remote Sensing Time Series"

_remotesensing, doi:10.3390/rs14225834_

Round 1

Reviewer 1 Report

This study assessed urban environmental change on the Google Earth Engine platform. This study is a successful application case. However, the manuscript is highly similar in structure to previous literature (the literature is located in the Appendix), including the title, secondary headings, etc. To address these issues, the authors should make appropriate adjustments prior to publication.

Author Response

The authors are very thankful to the reviewer for the suggestions to improve the quality and presentation of the manuscript. Following the comments made, the manuscript has been duly revised. The reviewer’s concerns are addressed in the following item-by-item form with sincere thanks from the authors. 1. Comment “This study assessed urban environmental change on the Google Earth Engine platform. This study is a successful application case. However, the manuscript is highly similar in structure to previous literature (the literature is located in the Appendix), including the title, secondary headings, etc. To address these issues, the authors should make appropriate adjustments prior to publication.” “Assessment of urban environmental change using multi-source remote sensing time series (2000–2016): A comparative analysis in selected megacities in Eurasia Linlin Lu, Qihao Weng, Huadong Guo, Suyun Feng, Qingting Li” Response: In the published article you introduced, an environmental index survey was conducted on 17 cities between 2000 and 2016 for cities that mostly included East Asia. In our research, the time period has been increased (from 2000 to 2019), and the environmental index has been examined for 32 major cities in the Middle East. We selected the Middle East region because of several reasons: 1) This region is highly under pressure because of climate change and drought; 2) Investigation of policy design by the government and its economic, environmental, and political impacts; 3) High urban sprawl; 4) Noticeable LULC change and high environmental degradation; 5) High level of Pollution because of low-price oil and dusty region. To make this research more comprehensive, the selected cities have much dispersion, so the results are highly generalisable. In addition, the growth of metropolises in the last two decades in the developing countries of the Middle East has been very high, and for this reason, this research is of great importance. The results of this research would greatly help increase research impact and social engagement and transfer information to people and politicians in this region. In this research, the Google Earth Engine platform was created to process satellite data at high speed. In the past years, this platform has become a powerful tool in various fields, such as environmental monitoring. The proposed approaches based on the GEE platform can be quickly developed and used in different studies by other researchers for other regions of the world. We also changed the title to make the similarity between our paper and the published papers the least amount.

Reviewer 2 Report

The article reports the Assessment of Urban Environmental Change Using Remote Sensing Time Series (2000–2019): A Comparative Analysis in the Middle Eastern Megacities. The GEE was applied while measuring PM 2.5, LST and NDVI. Finally, the authors investigate the improvement of CEI.

There are several shortcomings of this paper that should be considered.

1. The abstract should be more emphasized and should be concise.

2. Try not to use keywords directly from the title.  

3. Introduction should be consisting of 4 to 6 paragraphs, please reduced the paragraph in the introduction to concise sentences. I think this does not have a good impact on your manuscript from the reader.

4. The highlights of the paper do not include any innovative results and just have been focused on the results.

5. However, the methodology section is built not very well. How to calculate or collect LST and PM2.5 missing it?

6. What method have you used for image collection? Is it mean or median? please mention why?

7. Line: 241 MK trend analysis used but short time period Sen’s slope is better.

8. Can you use any ground truth verification of your research?                                                         
9. Line: 154-155: Silly grammatical mistake here.

10. Line: 254 starts with “figure 3 shows” please remove it.

11. The result discussion section is too long, some are repeated again.

Author Response

The authors would like to thank the reviewer for his/her precious time and invaluable comments. We have carefully addressed all the comments. The manuscript has been duly revised, and the reviewer’s concerns are addressed in the following item-by-item form.

  1. Comment “The abstract should be more emphasized and should be concise.”

Response: Thank you for this constructive suggestion. We revised the abstract section as you asked.

  1. Comment “Try not to use keywords directly from the title.”

Response: We updated keywords.

  1. Comment “Introduction should be consisting of 4 to 6 paragraphs, please reduced the paragraph in the introduction to concise sentences. I think this does not have a good impact on your manuscript from the reader.”

Response: We revised the introduction section and added some sentences to the introduction to highlight the importance of the paper.

  1. Comment “The highlights of the paper do not include any innovative results and just have been focused on the results.”

Response: We revised the paper in the introduction section to highlight the importance of this research.

  1. Comment “However, the methodology section is built not very well. How to calculate or collect LST and PM2.5 missing it?”

Response: The method of collection and the data used in this study are fully explained in the text in lines 181 to 223.

  1. Comment “What method have you used for image collection? Is it mean or median? please mention why?”

Response: In all sections, we assigned the mean value for image collection since the average value has the least difference with time series values.

  1. Comment “Line: 241 MK trend analysis used but short time period Sen’s slope is better.”

Response:  Thank you for your scrutiny. In this study, we extracted several parameters concerned with the MK test, but we used Sen’s Slope as the output of the MK test (Table. 4).

  1. Comment “Can you use any ground truth verification of your research?”

Response: The validity of the comprehensive CEI index in assessing and monitoring the environmental condition was examined in reputable journal papers such as He et al. (2017), and its capability was proven. In this study, we focused on introducing an automatic approach to investigate the conditions of the metropolises. This methodology can be implemented in other parts of the world with limited modifications. It is worth noting that the state of urban changes in four cities was examined visually and using satellite images during the last two decades. The results showed that this index was able to model the changes in environmental conditions well.

  1. Comment “Line: 154-155: Silly grammatical mistake here.”

Response: We edited the paper for such mistakes, and the changes are highlighted in the last version of the paper.

  1. Comment “Line: 254 starts with “figure 3 shows” please remove it.”

Response: We replaced this word and edited this part as you asked.

  1. Comment “The result discussion section is too long, some are repeated again.”

Response: Thank you for this constructive suggestion. We revised the paper as you asked.

Reviewer 3 Report

General suggestions: there are numerous typos and errors in the text. Please pay more attention and correct them; some are listed among the suggestions below. English is often redundant. Many sentences can and should be improved to make reading easier. The units of measurement in the international system (µg m-3) must be used. Check and correct figure numbering. It is recommended to report the data and trends identified by the scientific results. Naming "acceptable" or "unacceptable" conditions, needs accepted threshold values such as to identify different situations.

Specific suggestions: lines 22-28. Please be more succinct in mentioning environmental change

Introduction: Line 52 remove “according”; Line 53 “current”; Paragraph two Lines 57-72 appear redundant. Please, rewrite the whole paragraph. Line 77 Consider to change “spending a lot of money” with something as “time consuming”. Lines 79-80 consider to ad an example of your statement instead of just a couple of citations

Materials and methods: Paragraph 2.1. Please add a reference to table 1. Figure 1. Legend, consider to change World boundary with country boundary; Lines 137-138: Please refers only about datasets, methods and research topic are not required in this paragraph. Alternatively, move the sentence to the beginning of the paragraph as an introduction to the datasets used. 

Dataset: Please rephrase it. The introduction lines 135-141 should be shorter to give more emphasis to the characteristics of the datasets.

Methodology: Please report how the resolutions are standardized to 1 km for all datasets used. How has the size change of cities been treated over 20 years? Does the data presented and discussed derive at the average pixel over city boundaries?How was the data extracted? Line 298. Consider to change unacceptable with more scientific terms; Figure 5. It does not seem so informative. One understands a great variability of CEI from year to year, but it lacks a temporal trend as instead shown in Figure 3; Line 334, Figure 6 is missing. Line 350. Figure 7 is missing.

The presentation of results (lines 334-360) should be reformulated 1. by adding the figures. 2. without mentioning the figures and explaining more clearly what is being referred to; Lines 372-378 seems quite confusing. According to your results, are there or are there not environmental improvements in Tel Aviv and Alexandria? Figure 10-a., change with figure 11; Line 547, consider to change “better spatial accuracy” with “higher spatial resolution”.

The end of discussion lines 537-559 present only limiting and negative aspects of the study. 

It is recommended that the usefulness of the study be highlighted. Otherwise, a reader might wonder what was the point of all this work. Lines 563-567 the improvement provided by GEE are never discussed; Line 569 consider to change destruction with degradation or other words; Line 578 is it research instead of a review.

Author Response

The authors are very thankful to the reviewer for the suggestions to further improve the quality and presentation of the manuscript. Following the comments made, the manuscript has been duly revised. The reviewer’s concerns are addressed in the following item-by-item form with sincere thanks from the authors.

  1. Comment “General suggestions: there are numerous typos and errors in the text. Please pay more attention and correct them; some are listed among the suggestions below. English is often redundant. Many sentences can and should be improved to make reading easier. The units of measurement in the international system (µg m-3) must be used. Check and correct figure numbering. It is recommended to report the data and trends identified by the scientific results. Naming "acceptable" or "unacceptable" conditions, needs accepted threshold values such as to identify different situations.”

Response: All authors checked the manuscript carefully. One of our authors, who is a native English speaker, revised our paper for such mistakes, and the changes are highlighted in the last version of the paper.

  1. Comment “Specific suggestions: lines 22-28. Please be more succinct in mentioning environmental change”

Response: Thank you for your scrutiny. We did that.

  1. Comment “Introduction: Line 52 remove “according”; Line 53 “current”; Paragraph two Lines 57-72 appear redundant. Please, rewrite the whole paragraph. Line 77 Consider to change “spending a lot of money” with something as “time consuming”. Lines 79-80 consider to ad an example of your statement instead of just a couple of citations”

Response: We edited the paper for such mistakes, and the changes are highlighted in the last version of the paper.

  1. Comment “Materials and methods: Paragraph 2.1. Please add a reference to table 1. Figure 1. Legend, consider to change World boundary with country boundary; Lines 137-138: Please refers only about datasets, methods and research topic are not required in this paragraph. Alternatively, move the sentence to the beginning of the paragraph as an introduction to the datasets used.”

Response: We thank the reviewer for this observation; the problem has been fixed.

  1. Comment “Dataset: Please rephrase it. The introduction lines 135-141 should be shorter to give more emphasis to the characteristics of the datasets.”

Response: We revised the paper as you asked.

  1. Comment a) “Methodology: Please report how the resolutions are standardised to 1 km for all datasets used. b) How has the size change of cities been treated over 20 years? c) Does the data presented and discussed derive at the average pixel over city boundaries? d) How was the data extracted? e) Line 298. Consider to change unacceptable with more scientific terms; f) Figure 5. It does not seem so informative. g) One understands a great variability of CEI from year to year, but it lacks a temporal trend as instead shown in Figure 3; h) Line 334, Figure 6 is missing. Line 350. Figure 7 is missing.”

Response: Thank you, briefly:

  1. a) In this study, the spatial resolution for the PM5 and LST products is 1 km by default. The spatial resolution of the NDVI product is converted to 1 km using the resampling method (by default: nearest neighbour method) in the GEE platform;
  2. b) The border of the cities has been extracted according to the Google Map images in 2019;
  3. c) It depends upon the type of calculations. The is a pixel-based index and as mentioned in the manuscript, the is the mean of CEI annual time series in pixel i (Figures. 6-7). In the trend analysis section, calculations were based on the annual average of LST, VC, PM5 and CEI for each city boundary. The Mann-Kendall Test was applied to analyse the time series trend of each index (Table. 4). Finally, the pixel-based Mann-Kendall Test was investigated over all cities to represent the percentage of the urban area in which a significant trend for CEI index is observable (Figure. 9). In other words, in some part of results, we used mean values of pixels in time series (Figures. 6-7), and in other parts of results, such as trend analysis, we used the mean values of pixels within city borders to analyse the results (Table. 4);
  4. d) We assigned the mean value for image collection since the average value has the least difference with time series values;
  5. e) We revised the paper as you asked;
  6. f) In Figure 5, a box plot presents descriptive statistics and the annual average CEI from 2001 to 2019. The maximum long-term average of the CEI (from 2001 to 2019) is observed in the cities of Aleppo and Makkah. The results indicate that the environmental conditions in these areas are poor. In addition to these cities, the cities of Riyadh and Amman, Tehran, and Sharjah also show high CEI values. The lowest level of CEI was observed in Basra, Mosul, and Ankara. Interquartile ranges are high in the cities of Ajman, Basra, Jeddah, Mecca, Tehran, and Sharjah. This shows a great diversity in the environmental conditions in the studied metropolises during the study period. In addition, Figure 5 shows a skew in most studied cities. This deviation indicates a lack of stability in environmental conditions in the Middle East. Examples of cities exhibiting this instability are Dubai, Karaj, and Basra;
  7. g) We updated Figure 3 in the form that it also shows the trend of CEI;
  8. h) Thank you for your scrutiny. We did that.

  1. Comment a) “The presentation of results (lines 334-360) should be reformulated 1. by adding the figures. 2. without mentioning the figures and explaining more clearly what is being referred to; b) Lines 372-378 seems quite confusing. c) According to your results, are there or are there not environmental improvements in Tel Aviv and Alexandria? d) Figure 10-a., change with figure 11; Line 547, consider to change “better spatial accuracy” with “higher spatial resolution”.”

Response: Thank you, briefly:

  1. a) We revised this section;
  2. b) We thank the reviewer for this observation; the problem has been fixed;
  3. c) We revised this part of the result section to make it more understandable;
  4. d) Thank you for your scrutiny. We did that.

  1. Comment “The end of discussion lines 537-559 present only limiting and negative aspects of the study.”

Response: We removed this section to make our paper more understandable.

  1. Comment “It is recommended that the usefulness of the study be highlighted. Otherwise, a reader might wonder what was the point of all this work. Lines 563-567 the improvement provided by GEE are never discussed; Line 569 consider to change destruction with degradation or other words; Line 578 is it research instead of a review.”

Response: Thank you for the insightful comments. We edited the result section of the paper and added some sentences to this part.

Round 2

Reviewer 1 Report

The revised version can be accepted.

Reviewer 3 Report

I have checked carefully your work and you answered all my questions and modified the text according to what was asked. The paper has improved greatly and it is ready to be published by the journal. Thank you very much for what you have done.